# Stem Cell-Based Therapeutic Strategies for Premature Ovarian Insufficiency and Infertility: A Focus on Aging

**DOI:** 10.3390/cells11233713

**Published:** 2022-11-22

**Authors:** Ilyas Ali, Arshad Ahmed Padhiar, Ting Wang, Liangge He, Mingzhuang Chen, Shengda Wu, Yan Zhou, Guangqian Zhou

**Affiliations:** 1Department of Medical Cell Biology and Genetics, Guangdong Key Laboratory of Genomic Stability and Disease Prevention, Shenzhen Key Laboratory of Anti-Aging and Regenerative Medicine, and Shenzhen Engineering Laboratory of Regenerative Technologies for Orthopaedic Diseases, Health Sciences Center, Shenzhen University, Shenzhen 518060, China; 2Department of Ecology and Evolutionary Biology, University of Connecticut, Storrs, CT 06269-3043, USA; 3Lungene Biotech Ltd., Shenzhen, China; 4Guangdong Key Laboratory for Biomedical Measurements and Ultrasound Imaging, School of Biomedical Engineering, Shenzhen University Health Science Center, Shenzhen 518060, China; 5Senotherapeutics Ltd., Hangzhou, China

**Keywords:** premature ovarian insufficiency, infertility, pluripotent stem cells, germ cells, exosomes

## Abstract

Reproductive aging is on the rise globally and inseparable from the entire aging process. An extreme form of reproductive aging is premature ovarian insufficiency (POI), which to date has mostly been of idiopathic etiology, thus hampering further clinical applications and associated with enormous socioeconomic and personal costs. In the field of reproduction, the important functional role of inflammation-induced ovarian deterioration and therapeutic strategies to prevent ovarian aging and increase its function are current research hotspots. This review discusses the general pathophysiology and relative causes of POI and comprehensively describes the association between the aging features of POI and infertility. Next, various preclinical studies of stem cell therapies with potential for POI treatment and their molecular mechanisms are described, with particular emphasis on the use of human induced pluripotent stem cell (hiPSC) technology in the current scenario. Finally, the progress made in the development of hiPSC technology as a POI research tool for engineering more mature and functional organoids suitable as an alternative therapy to restore infertility provides new insights into therapeutic vulnerability, and perspectives on this exciting research on stem cells and the derived exosomes towards more effective POI diagnosis and treatment are also discussed.

## 1. Introduction

Premature ovarian insufficiency (POI), often referred to as premature ovarian failure (POF), is an endocrine disorder that describes a continuum of declining ovarian function, occurring in approximately every 1% of women ≥40 years of age and 0.1% of women ≥30 years of age [1,2]. In most cases, multiple factors contribute to the premature depletion of the primordial follicle pool [3]. Genetic abnormalities, autoimmunity, radiotherapy, chemotherapy, or surgery can all contribute to this disease [3,4,5,6,7]. The most common symptoms are menstrual irregularities, low-level estrogen, and high follicle-stimulating hormone (FSH); however, some POI patients have unexplained idiopathic causes [8]. POI at an early age has an undesirable impact on the reproductive system and may result in infertility. Furthermore, estrogen deficiency is linked to low bone density, cardiovascular diseases, sexual dysfunction, and high mental distress [9,10]. Though POI is a polygenic disease [11], the precise etiology and molecular mechanisms of POI remain unknown.

In recent years, the concept of aging as a disease has not prevailed in academia, but despite differences in preexisting concepts, it may not have the full symptoms of the specific disease, but still has inherent pathological features [12]. Likewise, aging has long been shown to be a major risk factor and more importantly associated with the faster progression of various health disorders [13,14]. Reproductive aging, which mostly develops over time, is a natural phenomenon [15]. Several studies have revealed the complexity and serendipitous uncertainty of the interconnectedness between reproductive aging and other organ aging [16]. Recently, numerous studies have described a strong connotation amongst inflammatory aging and POI [17,18]. In addition, ovarian biopsies of samples collected from POI patients showed lymphocytic infiltration and immune responses in the ovaries [19,20]. In the reproduction field, the significant functional role of inflammation-causing ovarian deterioration along with therapeutic approaches in order to prevent ovarian aging and increase their functionality are current research hotspots. However, further exploring the cause of POI and its association with aging needs to better understand the process of oocyte maturation and in what way the aging and inherited genetic defect are affecting it. Until now, limited research has been carried out on oocyte maturation because of the restrictions on researching and rarer approaches to the embryos of humans with inherited POI. Therefore, an effective cell model is crucial to deepen our knowledge of the pathogenesis and mechanisms of aging and to develop an innovative strategy for treating this prematurely ovarian disease.

Stem cell-based therapy, the core form of regenerative medicine, has been considered the greatest and most capable method in the present world of science and medicine field [21]. This cutting-edge technology has opened up limitless opportunities for transformative and potentially curative treatment options for the deadliest human illnesses [22]. Regenerative medicine is a growing and successful method of treatment and becoming the greatest therapeutic option in health care, with the specific goal of restoring and potentially substituting unhealthy cells, tissue, and organs and ultimately restoring normal function [23]. Luckily, regenerative medicine’s promise as a substitute for traditional drug treatments is gaining special attention due to the research community’s active efforts to investigate potential applicability for diverse disorders, comprising some life-threatening diseases. Recent studies reporting success in using stem cell therapy in patients have raised hopes that one day this regenerative strategy might be a treatment option for a variety of vexing diseases [24]. Indeed, the last several years have seen considerable development in clinical trials on stem cell therapy. Some of these trials have had a significant impact on various diseases [25]. To date, the most established stem cell therapy for POI treatment is mesenchymal stem cells (MSCs) [26,27]. However, despite a growing number of research findings on the success of stem cell-based therapies, there are still several experimental trials that have not received a complete regulatory endorsement to validate stem cell treatments.

The development of iPSC technology to reprogram somatic cells back to a pluripotent state followed by their subsequent differentiation into patient-specific cells offers a great opportunity to proficiently model and later on find suitable treatment options for certain intractable human diseases [28,29,30]. In addition to the feasibility of establishing disease models in petri dishes, other approaches, including in vitro development of iPSC-derived organs or animals combined with iPSC derivatives, may be superior strategies to obtain a more physiological environment for disease models [31,32,33]. By establishing an iPSC disease model, we not only gain insight into the disease molecular mechanisms but also create an exceptional platform for developing new drugs to achieve health and prevent or cure various aging-related diseases [34]. Therefore, the induction of iPSC differentiation into oocytes with the help of assisted reproductive technology (ART) allows access to advanced, corrected and safe transplantation materials that may bring important scientific, social and economic benefits, and with this assisted reproductive technology more and more couples can have children.

In this review, we discuss POI pathogenesis in general and describe its relationship to features of aging, and then further describe the current state of stem cell use in the diagnosis and treatment of POI, with a particular emphasis on the use of hiPSC technology, which could help to develop more mature and functional organoids suitable for future scientific research and alternative therapeutics in the field of regenerative medicine. We also discuss recent advances and current challenges in the field, while elucidating the development of human oocytes and germ cell-specific derivatives (exosomes) as a safe and effective therapeutic strategy to restore ovarian function in POI patients.

## 2. Premature Ovarian Insufficiency or Aging

Premature ovarian insufficiency, aging, or failure (POI, POA, or POF) refers to the loss of ovarian function in a woman at a younger age than the estimates for menopausal age [1,2]. POF is one of the leading but often overlooked cause of female infertility, in which a woman’s eggs are less likely to be fertilized, and may not even be fertilized. With premature ovarian failure condition, women are still having regular menstrual cycles as well as facing difficulty in conceiving, and is seen in young women, with about 10% experiencing fertility problems. This condition of early ovarian failure is referred by Center for Human Reproduction (CHR) researchers as a clinical term of premature ovarian aging (POA) [35], which in vitro fertilization (IVF) centers also refer to as primary ovarian insufficiency (POI).

Normally, the ovaries stop functioning and decline earlier than other organs in the body, which in turn ushers in the onset of systemic aging [36]. Due to modern society’s population structure and the delay of the age of first childbearing, ovarian aging and related issues are becoming increasingly serious. Along with the substantial extension of women’s life span in modern society, the asynchronous aging of the ovary and the whole body becomes in sharp conflict with pursuing better physical and psychological well-being in women at an advanced age. In the modern era, when infertility is regarded as one of the three most prevalent diseases affecting humans, infertility rates are as high as 15–20% in developed countries. In recent years, it has been reported that the rate of female infertility in China has increased from 3% to 15–18%, and approximately 40% are related to ovarian aging. Across the entire population of women of reproductive age, 1% to 3% of women reach pathological decline in ovarian function or pathological menopause before the age of 40, also known as premature ovarian failure (POF) [37,38]. Ovarian aging has emerged as a significant threat to women’s health in modern society as a result of rising rates of infertility and longevity. Therefore, delaying ovarian aging is the key to preserving fertility in older women, thereby improving their health and quality of life.

### 2.1. Pathophysiology of POI

The ovary is the most important female reproductive organ, performing both gametogenic and secretory functions [39,40]. A healthy ovary is necessary for proper functions and for the production of sex hormones [41], and this helps females regulate their hormonal growth, reproductive cycle, and menstruation throughout their lives. Ovarian dysregulation can have an impact on a woman’s physiological or reproductive functions. The primary constituents of follicles are granulosa cells (GCs) and oocytes, which are essential for ovarian functioning [42,43]. In addition to controlling oocyte growth through the secretion of growth factors and hormones, GCs are important in maintaining follicular evolution. Hormone receptor expression in GCs, for example, follicle-stimulating hormone receptors (FSHR), is critical in ovulation as well as in folliculogenesis. Interestingly, the number of operational oocytes demonstrates a female’s reproductive capacity [44]. During folliculogenesis, the mature oocytes accompanied by GCs can interact with relevant growth hormones/factors, such as bone morphogenetic proteins (BMPs) and FSH. In most cases, even though the cause of POI is unknown, expedited GCs and oocyte apoptosis obstruct follicle maturation, anomalies in follicle activation are key mechanisms of POI in general, and the hypothalamic–pituitary–ovarian axis regulates ovarian function [45]. The primary ovarian hormones that regulate folliculogenesis are FSH and LH. Furthermore, GCs secrete anti-Müllerian hormone (AMH) during the primordial follicle phase and initial phase of antral follicles. As a result, AMH plus estrogen has a negative effect on follicular growth and influences FSH levels. Once GCs are disrupted through chemotherapy or other treatments, estrogen and AMH levels decrease, causing FSH to rise and the follicular pool to be depleted [46,47]. POI is detected by a decrease in estrogen and AMH secretion and a rise in FSH secretion. Insights into the mechanisms underlying ovarian dysfunction and follicular pool depletion can thus aid in the identification and development of effective treatments for ovarian dysfunction [48,49]. The normal folliculogenesis and pathological symptoms of POF are presented in the Figure 1.

### 2.2. Causes of POI

POI is accepted in clinical practice in females under the age of 40 who show sex hormone deficiency, amenorrhea, and serum FSH level greater than 40 IU/L [50]. POI results in low levels of progesterone and estrogen. If untreated, it increases the risk of cardiovascular disease, osteoporosis, and cognitive issues. The beginning of cycle irregularity and the final cessation of menstruation is determined by the age-related decrease in follicle numbers. Natural sterility and the gradual decline in fertility are both caused by the parallel decline in oocyte quality. The decrease in negative feedback from ovarian factors at the hypothalamic–pituitary unit is the primary cause of endocrine changes. FSH levels rise first as the number of antral follicles decreases with age, then there are stages of overt cycle irregularity. Decreased levels of AMH best illustrate the gradual shrinkage of the antral follicle cohort.

Pathophysiological ovary conditions can result in follicle destruction or follicular abnormalities in cases of iatrogenic treatments, autoimmunity, genetic abnormalities, and environmental issues, and cause metabolic abnormality and infertility. Genetic factors contribute to approximately 20–25% of POI cases, where chromosomal abnormalities and genetic mutations are responsible. Multiple relevant genes have been implicated in ancestral POF, and about 14% of these cases have a positive POI genetic history [51]. Atypical chromosomal organization and mutations in particular genes linked to ovarian function or metabolic control are two examples of the genetic reasons for follicular dysfunction [52]. The DNA may also be damaged by chemotherapy and radiation therapy, which might eventually affect ovarian function by inducing apoptosis in GCs and loss of preantral follicles [53,54]. Follicular atresia is brought on by GC malfunction, and has been identified as the primary factor in POI [55]. In individuals with biochemical POI, it was shown that granulosa cell-associated transcript 1 is downregulated in GCs, demonstrating a unique lncRNA approach that promotes epigenetic control of GC activity and aids in the etiology of POI [36]. Furthermore, POI has been associated with many autoimmune diseases [8]. The autoimmune attack’s damage to ovarian function might be the underlying pathogenic factor. Even so, the exact function of the autoimmune response in ovarian dysfunction requires further investigation. However, a majority of POI cases are idiopathic [53], which encourages future research to fully comprehend this condition and investigate novel solutions.

In cases of idiopathic POI, the understanding of possible causative genetic defects and the relationship of aging with POI remains to be elucidated. Recently, numerous studies have described a strong connotation between inflammatory aging and POI [17,18]. In addition, ovarian biopsies of samples collected from POI patients showed lymphocytic infiltration and immune responses in the ovaries [19,20]. However, exploring the causes of POI and its association with aging needs to gain an understanding of the process of oocyte maturation and in what way aging is affecting it. Until now, limited research has been carried out on oocyte maturation because of research restrictions and rarer approaches to the embryos of humans with inherited POI. A variety of diseases are linked to POI, even in instances with a known origin, showing heterogeneity of this entity, and thus the identification of the exact causative defect is essential. This fact emphasizes the necessity of creating several ways to enhance the clinical care of these patients, as well as the significance of choosing the appropriate demographic of POI patients who can profit from each strategy. As a result, it is critical to investigate efficient treatment for managing POF and associated complications. Specific causes of POF require specific treatments according to the guidelines of the European Society of Human Reproduction and Embryology (ESHRE). ESHRE recommends HRT for POI treatment under conditions with hypoestrogenic symptoms [56]. Despite HRT being the standard clinical treatment for POI, it fails to restore ovarian function completely. Other treatment options, such as in vitro follicle activation, oocyte cryopreservation, and ovarian tissue transplantation, have also been investigated [57]. However, these methods have not yet been used in medical applications. The reasons are inefficient follicle activation and technical, ethical, and administrative constraints. Rapid advancements in regenerative medicine hold great potential to restore ovarian function using stem cells. Figure 2 shows the likely causes of and available treatments for POI.

### 2.3. Premature Ovarian Aging and Infertility

Female fertility is negatively affected by premature ovarian failure due to insufficient egg numbers and poor egg quality [16]. A few low-quality eggs can markedly reduce the fertility rate of women in two ways: lowered fertility, or in cases of pregnancy, miscarriages usually occur. Unluckily, egg quality is inversely proportional to egg quantity [17,18]. As a result, females with untreated reduced ovarian reserve have the highest rates of miscarriages of any infertility diagnosis, as approximately 95% of quality embryos come from eggs, making miscarriages more likely due to poor embryo quality. However, deducing the exact mechanism of premature aging requires better understanding of the process of oocyte maturation and in what way aging and inherited genetic defects are affecting it. Limited research has been carried out on oocyte maturation because of the restrictions imposed on such research and rarer approaches to the embryos of women with inherited POI. Therefore, there is an urgent need for an effective POI model to deepen our knowledge of the pathogenesis and mechanisms of premature aging and to develop an innovative strategy for treating this ovarian aging disease and preventing infertility.

## 3. Aging Hallmarks of Premature Ovarian Insufficiency (POI)

The underlying mechanisms of ovarian aging remain poorly understood, especially since it is a complex biological process in which many factors interact internally and externally. The combination of internal and external elements during the body’s degeneration stage causes the long and complicated biological process of natural aging in humans. Structures deteriorate with age, the internal environment becomes imbalanced, function declines, and adaptation, resilience, and resistance are lost [58]. The study of aging is entering a new phase of inquiry due to the rapid advancement of technology, although the precise process behind aging has not yet been fully understood. Inflammation has been linked to aging in a large number of reports. Aging is followed by a persistent and developing proinflammatory condition in the body tissue and organs [59,60,61]. Figure 3 shows the important aging hallmarks of premature ovarian insufficiency.

### 3.1. Inflammatory Aging and POI

#### 3.1.1. Oxidative Stress and Inflammatory Aging

Oxidative stress (OS) is caused by disproportionate amounts of oxidants and antioxidants in the body. This is oxidative in nature and results in inflammatory infiltration of neutrophils. An excessive amount of reactive oxygen species (ROS) generated by chronic inflammation, such as oxygen free radicals, are key factors of body inflammatory aging [62,63]. Since oxidative stress and aging are closely related, Sohal and Weindruch [64] developed the idea of oxidative stress in the 1990s after pointing out the flaws of the free radical hypothesis. Ottaviani and Franceschi [65] looked at a variety of animals, including invertebrates and humans, and discovered that immune stress inflammation constitutes the body’s defensive network, demonstrating that stress is one of the factors that contribute to inflammatory aging. Antioxidants have effectively been used to lessen oxidative stress-related damage and increase human survival, supporting the theory of oxidative stress inflammation [66,67].

#### 3.1.2. Oxidative Stress and Inflammatory Aging

An essential component of inflammatory aging is a highly proinflammatory state brought on by the overexpression of inflammatory factors in the body. Numerous investigations have demonstrated that aging organs have considerably greater blood inflammatory markers such as PGE2, IL-8, IL-6, and TNF-α [68,69,70]. Salvioli et al. [71] investigated healthy people of various ages and those with aging-related conditions, e.g., Alzheimer’s and type 2 diabetes, and they came to the conclusion that proinflammatory cytokine levels were significantly correlated with aging.

#### 3.1.3. Inflammatory Aging Causes POI

Inflammatory variables may be a major contributing factor to POI, according to many local and overseas studies that suggest that IL-6 and TNF-α may affect ovarian function [72,73]. Exogenous factors, such as surgery, drugs, and the environment, and endogenous factors, such as chromosomal linkage defects [74,75], autoimmune reactions [76], physiological stress [77], genetic predilection [7,78], and some enzymatic deficiencies, are involved in ovarian changes. Different research data have shown that ovarian functions are affected by TNF-α and IL-6 [73,79], which shows that inflammatory factors are the major reason for POI. The present review explores the association between POI and inflammatory aging based on the following points.

Recently, there has been tremendous interest in the role of inflammatory aging in ovarian disease. During follicular rupture, IL-1 and TNF-α are the major cytokines involved, and this process is considered an inflammatory reaction. In recent years, studies have shown that in the female reproductive tract, the abnormal manifestation of xanthogranulomatous inflammation indicates POI, suggesting that inflammatory aging is one of the main causes of POI [79]. Immune responses mediated by T helper cells (such as Th1 cells) are positively correlated with inflammatory responses. Through experimental induction of autoimmune ovarian inflammation by inhibin-alpha, CD4 (+) Th1 T cells are initiated, inhibin-alpha neutralizing Abs are produced as a result of B cell stimulation, directly initiating POI and disease metastasis in local receptors [80]. During inflammatory and immune responses, different molecules are involved that are under the control of NF-κB signaling, comprising IL-2, IL-1β, IL-6, TNF-α, and colony-stimulating factors. Anti-inflammatory factors such as zinc finger protein A20 and HO-1 (heme oxygenase 1) are controlled by NF-κB signaling. During transcription, NF-κB induces all genetic factors that are involved in immune responses and inflammation. Inflammation markers, e.g., IL-8 and IL-6, and mRNA expression of anti-inflammatory cytokines (IL-10) reduce during proinflammatory cytokine activation [81].

Scientists have discovered that NF-κB activity increases with aging in the mouse brain and has an insignificant role in the young mouse hypothalamus. This shows that the high level of inflammatory cytokines and lower concentration of anti-inflammatory cytokines have a crucial effect on POF. Recent research has reported that the inflammatory factor TNF-α is higher in patients with POF compared with normal conditions. Wang et.al. [82] found that anti-zona pellucid antibodies (AzpAb) levels are considerably higher in POI patients than normal controls, IL-2 and TNF-α concentrations reduced notably, and INF-γ levels increased. The level of IL-2 and TNF-α in POI patient serum was found to be markedly reduced compared to the controlled group because of the TNF-α production by immune cells such as lymphocytes and granulocytes, while patients with POI had atrophied ovarian and granulocyte cells, which probably decreased the level of serum and the inflammatory factors. Sundaresan et al. [83] found in birds that the concentration of chemokines and inflammatory cytokines was higher in POF, showing a positive association with inflammatory aging.

With the repair of ovarian function, the number of inflammatory factors gradually decreases. According to new research, synovial mesenchymal stem cells (SMSCs) show a positive effect by repairing damaged ovarian follicles [84]. In a recent study, the application of SMSCs to treat ovarian injury in mice showed a significant reduction in proinflammatory cytokine levels. During POI, the levels of some inflammatory factors increased, and with the recovery of damaged ovarian function, the levels of most inflammatory factors decreased, indicating that POI is positively correlated with inflammatory aging.

### 3.2. Mitochondrial Damage and Ovarian Aging

Mitochondria are the oocyte’s most abundant organelles and are the main source of energy for fertilization and maintenance of embryogenesis. The important function of mitochondria in oocyte meiosis mainly includes cell maturation, spindle assembly, and chromosome segregation [85]. Thus, changes in the distribution and quantity of mitochondrial and mtDNA sequences have a close association with oocyte quality and possess significant implications for embryonic development [86]. According to the constructed model of secondary oocyte OS injury, both mitochondrial membrane potential and ATP levels were found to decrease and were followed by spindle damage [87]. The cytochrome C oxidase 1 gene located in mitochondria is a key regulator of mitochondrial OS, and its mutation was reported to increase in a patient with POI [88]. Increased OS and increased susceptibility of oocytes to OS cause chromosomal abnormalities, spindle instability, shortened telomeres, and reduced developmental capacity of senescent oocytes [89].

Dysfunction of mtDNA, increased oxidative damage, altered membrane potential, and insufficiency of mitochondrial biosynthesis or clearance due to mitochondrial dysfunction are all factors that contribute to ovarian aging [90]. Examples of mtDNA failures that reduce mtDNA contents include oxidative damage, strand breaks, and point mutations. It has been reported that mtDNA in POI patients is much lower than in healthy women of childbearing potential [91]. In addition, the loss of histone protection and DNA repair enzymes makes mtDNA susceptible to mutation [92]. According to research, a single point mutation in the mitochondrial DNA has a significant impact on ROS production and mitochondria proteostasis shortens telomeres [93]. Mice given mutant mtDNA polymerase gamma (POLG) age rapidly [94]. ROS levels are significantly increased in POI inhabitants [95]. Age-induced mtDNA mutation and energy deprivation are caused by excessive accumulation of ROS [96]. Then, mtDNA mutations make ROS production worse. Apoptosis may result from this vicious cycle of self-amplification and annihilation. Furthermore, excess ROS production may be too much for the cellular antioxidant defense capacity and causes OS and ovarian aging [97].

Oocyte aging is also influenced by disturbances in the dynamics of mitochondria, for example, mitochondrial fusion, modifications in the metabolism of mitochondrial, and inconsistencies in calcium homeostasis [96]. Oocytes missing mitofusin 2 (MFN2) led to female infertility because it is an important protein causing mitochondria fusion [98]. The essential component for preserving oocyte quality is mitochondrial Drp1 (fission factor dynamin-related protein 1). Drp1 knockout has been linked to follicular dysplasia and ovulation issues [99]. Additionally, the absence of mitochondrial proteases may cause disorders linked to mitochondria and hasten oocyte aging [100].

Age-linked alterations in oxidative stress and mitochondrial function-related gene expression have been reported in mice oocytes [87]. POLG1 gene mutation leads to increased mitochondrial DNA mutation and has been reported to cause premature aging in mice. In addition, patients with POLG1 mutations exhibit phenotypes of premature menopause and SNPs associated with the POLG1 gene are associated with GWAS-based age at natural menopause [51,101]. Moreover, a significant reduction in mitochondrial DNA copy number was found in oocytes and peripheral blood cells in POI patients. An additional mitochondrial protein, histidyl-tRNA synthetase 2 mutation in Perrault syndrome patients display a POF phenotype, endorses the significance of mitochondrial functionality of the ovary [102,103]. Hence, these results suggest that mitochondrial dynamic dysregulation leads to extreme OS and the start of apoptosis, thereby accelerating follicle depletion and causing ovarian senescence.

### 3.3. Premature Ovarian Aging and Senescence

Ovarian aging is not only significant in itself but has also received attention for its general biological relationship to senescence [104]. A vital feature of aging is the peculiarity among germline immortality and death of somatic cells and tissue in higher animals. Ovaries containing female germ cells, due to which the contribution of females to immortality of the germline is compromised [105]. It is already known that individual oocytes can and do age, and therefore ovarian aging performs an important role in starting or speeding other cascades of aging changes [106]. In order to better recognize how premature ovary acclimates to the overall biology of senescence, it is imperative to elucidate the causes of aging, examine possible mechanisms of aging in general, and enquire if there is any abnormality in ovary aging and their association with POF [35]. At many levels, from the progression of human life history to the biochemical and cellular principles of aging and longevity, the study of ovarian aging intersects with our new understanding of the biology of aging in general.

Premature menopause occurs earlier than 45 years of age, and POI occurs mostly at age less than 40 years due to natural senescence and therefore represents a continuum of phenotypes, and more interestingly several POI patients were also identified in presentation of premature menopause [107,108]. Among the early menopause susceptibility candidate loci identified by GWAS, one repeat gene is POLG1, which is essential for reducing mitochondrial DNA mutations [109]. A huge number of loci are also situated close to genes in DNA repair pathways and have been identified as candidate POFs. Additionally, a few POI genes, such as a number of receptor genes (such as FSHR, LHCGR, TGFBR1, and IGF2R) and a number of ligand genes (such as IGF1, TNF, and FSHB), are also linked to the beginning of female menopause [6,110]. Reduced expression of critical genes discovered by mutations in POI patients may, at least in part, explain ovarian senescence. Contrarily, due to the lack of ovarian samples, particularly oocytes, it is very difficult to evaluate genes that indicate diminished function in middle-aged females. This is in contrast to ovarian infertility and subfertility genes.

### 3.4. Telomere Shortening and DNA Damage in POI

Damaged DNA may be repaired by the normal body’s DNA repair system, but as we age, DNA damage accumulates and ultimately results in cell death. DNA damage causes the inflammatory aging of the body: when stem cells and stromal fibroblasts differentiate, this damages the multi-shell cytokine network and causes proinflammatory cytokines to be overexpressed [111].

Telomeres have been reported to perform an essential job against stress conditions in aging by regulating cellular responses and growth as antidotes to DNA damage and cell division [112]. In living beings, due to the extended interval between oocyte formation and the occurrence of ovulation, the oocyte telomeres decline with the passage of time. In age-related reproductive failure, a dysfunction may cause impaired spindle assembly formation and failed chromosomal segregation [113]. The mean leukocyte telomere length of an older female who give birth to a child with Down syndrome was shorter than that of age-matched females who gave birth to children with normal karyotypes [114]. In addition, reduced activity and shortening of telomerase were reported in granulosa cells of POI [115,116]. Hence, these results represent the telomerase regulatory mechanism in ovarian telomere homeostasis and ovarian aging.

### 3.5. Apoptosis

Apoptosis is also extensively reported in the process of ovarian aging. Ovarian apoptosis can lead to follicular atresia or degeneration extensively, and has been shown to be one of the central figures in the ovarian aging mechanism. Extrinsic and intrinsic pathways, as well as endoplasmic reticulum stress, have all been implicated in OS’s ability to induce ovarian cell apoptosis. Apoptosis in oocytes is directly associated with impaired germ cells [86]. Granulosa cell apoptosis causes nutritional deficiency and metabolic problems in the ovarian microenvironment, resulting in a decline in ovarian function. Additionally, it has been reported that OS is primarily responsible for the apoptosis of germline stem cells (FGSCs), thereby reducing their stemness and proliferative capability. The ovary loses the capacity to replenish the primordial follicles required for oocyte production, and eventually the ovarian reserve is gone [117]. The generated cell debris resulting from apoptosis can go on to affect the ovarian microenvironment, increasing the free DNA level in the follicular fluid and intracellular ROS production, which further aggravates apoptosis [118].

Apoptosis and the incidence of POI are tightly connected [119]. The preservation of the ovarian and follicular microenvironment’s homeostasis is necessary for the development of oocytes. In all phases of oocyte production and following ovulation, apoptosis is crucial for the removal of germ cells. A majority of germ cells in the ovary die as a result of follicular atresia, but only 1% of them grow into oogonia. As a result, when the apoptosis process is out of balance, it results in follicular atresia without oocyte maturation and early depletion of the ovarian reserve depletion [118]. To aid in the development and maturation of oocytes, granular cells release a range of chemicals. Additionally, the oocytes produce a range of cytokines that interact with granulosa cells’ surface receptors to encourage their growth and differentiation. The contacts between oocytes and granulosa cells are connected to one another in various ways as the follicles develop [120], and the gap junctions were essential for the communication of information and the movement of materials [121]. Granulosa cells’ interactions with oocytes control follicular maturation, ovulation, and fertilization. The signaling pathways involved are intricate and intertwined, with the main players being cyclic adenosine monophosphate (cAMP) and PI3K/Akt. Oocyte apoptosis is brought on by the same process that causes granulosa cells to die [122,123,124,125]. Due to the acceleration of follicular atresia caused by these occurrences, POF results. Apoptosis exhibits unique regulation patterns and variables depending on the physiological milieu and follicular development of the ovary.

Recent research revealed that the onset and progression of POF may be significantly influenced by OS. By inducing oocyte apoptosis, Wang et al. [126] showed that diisooctyl phthalate and its metabolites impede the development of antral follicles in mouse ovary. According to Sobinoff et al. [127], smoking can induce antral follicle oocytes to apoptose through OS, which results in defective follicles. By subcutaneously injecting tripterygium glycosides, Ma et al. [128] created a mouse model of early-onset ovarian insufficiency. In addition, they measured the levels of anti-Mullerian hormone (AMH), glutathione peroxide (GSH-Px), malondialdehyde (MDA) and superoxide dismutase (SOD) in the serum and ovarian homogenate of mice in the control and POF groups. Notably, the levels of SOD and GSH-Px in the serum and ovarian homogenate of the POF group’s mice were much lower than in the control group. According to Tokmak et al. [129], iNOS, MPO, and TOS levels in the blood of POF patients were greater than those in healthy women of reproductive age, indicating that OS was implicated in POF. Salminen et al. [130] discovered that as people age, their ability to clear defective proteins and mitochondria from the body increases, which raises ROS and oxidative stress levels. ROS causes a number of inflammatory responses, including an increase in the production of IL-1 and IL-18, by activating the NOD-like receptor 3 (NLRP3). As a result of an insufficient autophagic process, these cytokines further hasten the aging process by causing inflammation [131]. Using Chinese hamster ovary cells, a model ovarian cell with excessive autophagy was successfully created, serving as a research tool for in vitro POI studies [132]. Melatonin treatment successfully halted the excessive autophagy-induced follicular atresia of ovarian granulosa cells, which is thought to be one of the pathogenesis of POI and reduces mitochondrial and ovarian reserve functionality. This suggests that premature ovarian failure (POF) is associated with an increase in autophagy, which may be caused by an increase in apoptosis.

There are obvious limitations to the existing information and understanding of female reproductive aging. We are aware that ovarian changes play a major role in this process, which ends with menstrual cyclicity because of the ongoing loss of follicles and their eventual depletion. Alterations in ovarian feedback are considered responsible for changes in the neuroendocrine control of cyclic ovarian function. Antral follicle counts and AMH currently provide the most accurate representation of periosteal follicle loss, though these markers may not completely capture the entire decline. In addition to the decrease in quantity of oocytes, the parallel decline in oocyte competence and its effect on female fertility with age have been recognized. Aneuploid embryos are becoming more common as a result of the quality loss in oocytes, but the precise molecular biology mechanisms are still poorly understood. There is a great deal of variation among women when it comes to both the quantity and quality of ovarian decline aspects of aging. This variation suggests that many women face a clearly shorter fertility life expectancy.

## 4. Stem Cell-Based Strategies for POI and Infertility Treatment

Stem cells are unique in that they are capable of self-renewal and can differentiate into particular tissue types in response to their environment and signals. Stem cells can be loosely classified as pluripotent or multipotent based on their ability to differentiate [133]. ESCs, which originate from the inner blastocyst cell mass and have an unlimited capacity to differentiate into a wide variety of cell types, are the most important pluripotent stem cells [53]. Endoderm, mesoderm, and ectoderm-derived cells can all be grown from pluripotent stem cells. ESCs are rarely used in research despite their limitless potential due to ethical concerns and the risk of cancer, including the development of teratomas. People who oppose disrupting the embryo view the derivation of ESCs as immoral because the process of separating ESCs involves rupturing the blastocyst [134]. There are substitutes to employing ESCs, such as iPSCs and MSCs, which are mostly employed in research, taking into account ethical considerations. In order to take advantage of the increased potency of ESCs, iPSCs are utilized as an alternative source of pluripotent stem cells. iPSCs are synthetic stem cells that are produced by reprogramming low-potency, specialized cells like fibroblasts. As a result, multipotent cells can be utilized without posing any ethical concerns. iPSCs are used as an alternate source of pluripotent stem cells to take advantage of ESCs’ enhanced potency [134]. Mesenchymal stem cells (MSCs) are multipotent adult stem cells, that are frequently utilized in medical research and treatment despite the fact that they originate from the mesoderm, their capacity to transform into a variety of tissue and cell types, including chondrocytes, osteocytes, adipocytes, hepatocytes, and even neurons [135]. MSCs can also be extracted from the skin, adipose tissue, amniotic fluid, the placenta, the umbilical cord, and bone marrow [136]. MSCs’ advantages include chemotaxis during wound healing, minimal immunogenicity, limited immunomodulatory action, and no ethical implications. However, the ability to self-renew is limited by donor age and invasive collection methods like liposuction and bone marrow aspiration [137]. Numerous researchers have defined and evaluated the therapeutic potential of various stem cells in a variety of animal models of degenerative diseases [138].

### 4.1. Mechanisms of Stem Cell Therapy in POI

The therapeutic effects of stem cells are mediated via homing, differentiation, and paracrine activation. Numerous circumstances cause stem cells to voluntarily move to the damaged ovary to adhere and multiply. According to recent research, paracrine processes may be responsible for the curative benefit of stem cell transplantation [139]. In order to affect neighboring cells, surrounding cells secrete a variety of physiologically active chemicals, such as cytokines, growth factors, regulatory factors, and signal peptides. As shown in Figure 4, this procedure enhances the health of injured ovaries through immunological modulation, angiogenesis, antiapoptosis, antifibrosis, and anti-inflammation.

Further analysis of paracrine signals reveals that exosome secretion by stem cells serves to facilitate their activity. Exosomes are extracellular vesicles (30–140 nm in size) that transport proteins, mRNA, and microRNA [140]. These vesicles enable cell-to-cell communication through targeted cell internalization, ligand–receptor contact, or lipid membrane fusion [141]. Exosomes may start repair and regeneration processes based on their parent cells, which restore vital cellular activities and preserves tissue homeostasis. Another new technique created recently is stem cell-mediated mitochondrial transfer. Pro-inflammatory cytokines that cause skeletal reorganization in stem cells create tunnel nanotubes that allow mitochondria to be transported from stem cells to neighboring cells. In addition, it was recently discovered that mitochondria derived from stem cells can be transferred into oocytes, potentially restoring oocyte quality and embryonic development [142,143].

### 4.2. Preclinical Studies of Stem Cell Therapy for POI

Recent research shows that stem cell transplantation may have therapeutic promise since it enhances ovarian function by boosting the production of hormones that are connected to it and by decreasing GC apoptosis [144]. A number of variables, including folliculogenesis, GC apoptotic rate, vascular development, pregnancy rate, and hormone level modulation, can be used to quantify the therapeutic benefits of stem cells [145]. The stem cell transplantation can speed up oogenesis or provide favorable surroundings, which can increase folliculogenesis. Theca cells and GCs are known to become stiffer when produced from adipose tissue (ADSCs), creating the ideal conditions for folliculogenesis [146]. Peripheral blood mononuclear cell (PBMC) populations contain a number of multipotent progenitor cells, including stem cells, that are capable of regenerating tissue and restoring organ function. Platelet-rich plasma (PRP) and PBMCs had a synergistic effect on restoring folliculogenesis when transplanted into a POI mouse that had been induced by cyclophosphamide (CTX) [147]. In addition, human endothelial progenitor cell (hEPC) transplantation reestablishes control over inflammation, apoptosis, and endoplasmic reticulum (ER) stress, mitigates the effects of aging on reproductive health, and restores the ability to produce embryos [148].

#### 4.2.1. Mesenchymal Stem Cells

Mesenchymal stem cells, which are multipotent stem cells, which can originate from adipose tissue, umbilical cord blood, or bone, have the advantages of being easily accessible and lacking immunogenicity [149]. Researchers treated POI in mice using umbilical cord-derived mesenchymal stem cells (UCMSCs). They reported improved ovarian function, a decrease in cumulus cell death, and a rise in sex hormone levels. In the end, they made a comparison between the RNA expression in the treated group, the wild-type control group, and the POF model and reported similarities between the wild-type group and the treated group using UCMSCs [150]. In addition, it has been demonstrated that ovarian reserve function and follicle quantity were enhanced when hUCMSCs were transplanted into rats on days 14, 21, and 28. FSH levels decreased and AMH as well as E2 levels increased. This study also demonstrated that hepatocyte growth factor (HGF), vascular endothelial growth factor (VEGF), and insulin-like growth factor-1 (IGF-1) can be released by hUCMSCs [151]. When MSCs from the bone marrow of male rabbits were transplanted into POF-induced rabbits, studies revealed this boosted VEGF, decreased FSH, and increased the number of follicles with normal structure [152]. It was further demonstrated in mice that human endometrial mesenchymal stem cells (EnSCs) taken from menstrual blood improved the estrous cycle and restored fertility. Additionally, the pool of germline stem cells (GSCs) was less depleted after EnSCs were transplanted into a rat with a damaged ovary [153]. In 2017, another group of researchers transplanted mice with bone marrow-derived human mesenchymal stem cells (BMSCs) and observed an increase in body weight and ovarian weight following the transplantation of BMSCs, which resulted in the restoration of folliculogenesis and ovarian hormone production [154]. Adipose-derived stem cells (ADSCs) have also been utilized in mice with chemotherapy-induced ovarian damage. The number and health of follicles were counted and evaluated after the ovaries were removed and ADSCs were transplanted either directly into the bilateral ovaries or via intravenous injection a week or a month later. Improvements in ovarian function, ovulation, and follicle population were found in this study [155]. The application of human menstrual blood stem cells (hMensSCs), human menstrual blood stem cells (hMensSCs), which have been shown to lessen the apoptosis of granulosa cells and the fibrosis of the ovarian interstitium, have also been studied for their application. hMensSCs produced higher levels of ovarian markers, an increased ovarian weight, a higher plasma E2 level, and a normal number of follicles after being transplanted into the mice [156]. Additionally, it has been investigated how hMenSCs release FGF2 to further reduce granulosa cell apoptosis and fibrosis of the ovarian interstitium and protect injured ovaries. MenSCs increased the number of follicles and improved the microenvironment of the ovary by releasing FGF2 [157]. Although these findings offer promising preliminary evidence for the therapeutic potential of MSCs for POF infertility, additional research is required to determine their safety and efficacy in clinical settings.

#### 4.2.2. Bone Marrow Stromal Cells

Multipotent adult stem cells of the BMSC type can be identified in the microenvironment of the bone marrow. Under certain conditions, these cells can transform into a variety of cell types, including adipocytes, cartilage, and bone [158]. BMSCs lack immunogenicity because they do not express major histocompatibility complex class II molecules and produce several anti-inflammatory proteins [159]. Additionally, BMSCs may be candidates for ovarian tissue repair due to their propensity to travel to wounded areas and establish homes there [160]. They are actively involved in the healing of tissue injury by controlling the immune response and improving the function of associated cells through a variety of cytokine profiles [161,162]. Although larger population sizes are required for further trials to ensure clinical efficacy and applicability, these preliminary findings suggest that MSCs can restore fertility.

#### 4.2.3. Adipose-Derived Mesenchymal Stem Cells

Adipose-derived mesenchymal stem cells (ADSC) are another type of multipotent MSC found in adipose tissue [163]. Compared to harvesting these cells from other sources like BMSCs, using collagenase digestion and centrifugal density gradient separation on processed lipoaspirate is less unpleasant [164]. Early applications of ADSCs in regenerative medicine focused primarily on their ability to differentiate between endoderm, ectoderm, and mesoderm lineages [165]. Injured tissue can repair itself thanks to their ability to have paracrine effects on a variety of cytokines, chemokines, and growth factors [166]. However, in order for ASDCs to be utilized as future therapeutics, a number of preclinical studies may be required to ensure their safety and clinical compatibility.

#### 4.2.4. Extraembryonic Stem Cells

Extraembryonic layer-derived amniotic fluid stem cell (AFSCs) is a multipotent population and more rapidly proliferating than mesenchymal stem cells and bears markers of both embryonic and adult stem cells [167]. It has been demonstrated that transplanting AFSCs from transgenic mice prevented follicle atresia and preserved healthy follicles [168]. In 2013, granulosa cell differentiation was demonstrated using human amniotic epithelial cells (hAECs). As a result, the treated mouse’s ovaries showed signs of the ovarian function marker anti-Müllerian hormone [169]. Additionally, studies demonstrated that miR-21 prevented granulosa cells from dying by apoptosis by targeting the proteins phosphatase and tensin homolog (PTEN) and programmed cell death 4 (PDCD4). As a result, increased miR-21 expression MSCs decreased GC and follicle apoptosis while increasing E2 and decreasing FSH [170]. miR-10a, which is produced from stem cells in amniotic fluid, has anti-apoptotic properties. These microRNAs’ target genes are necessary for apoptosis. In this study, researchers discovered that exosomes produced from AFSC prevented ovarian follicle atresia in mice with chemotherapy (CTx)-induced POF in vitro and that downregulating miR-10 and miR-146 reduced the antiapoptotic effects on damaged GCs [171]. They found that cytokines in the hAEC-CM play a role in such biological processes as angiogenesis, immune response, cell cycle, and apoptosis after injecting human amniotic epithelial cells (hAECs) or hAEC-conditioned medium (hAEC-CM) into the unilateral ovary of a POF mouse [172].

Various subtypes of stem cells can be separated during the prenatal stage. The placenta, umbilical cord, and amnion are all sources of MSCs. The term “human amnion-derived MSCs” (hAD-MSCs) is frequently used to refer to the stem cells that can be found in amniotic fluid as well as human amniotic epithelial cells [173,174,175]. MSCs obtained from the umbilical cord have a very low initial immunogenicity, making them suitable for allogeneic treatment [176]. Compared to perivascular stem cells (PSCs) from umbilical veins or MSCs produced from Wharton’s jelly, PSCs from umbilical arteries are more likely to express the Notch Ligand Jagged 1 than MSCs from the umbilical cord [177]. Rebuilding ovarian function frequently involves the use of human chorionic plate-derived MSCs (hCMSCs), also known as human placenta-derived MSCs (hPMSCs). It has been demonstrated that hPMSCs secrete a variety of cell factors, including IL-6, IL-8, and IL-10, granulocyte colony-stimulating factor, and chemokine (C-C motif) ligand 5 [178]. Even though there are only a few studies on this application, they show promising results and should be considered for more in-depth research and eventual clinical application.

#### 4.2.5. Ovarian Stem Cells

The idea that an adult mammalian ovary has a set number of oocytes has been debunked by recent investigations. The expansion and maturation of ovarian germline stem cells (GSCs) may open up new treatment options for POF. Despite the widespread recognition of ovarian GSCs in non-mammalian model animals, recent research in mice, rats, and humans has found that ovarian GSCs exist, which will help with infertility therapy and may one-day help with POF disease [179]. Putative ovarian mesenchymal stem cells (PO-MSCs) have also been found in ovarian cortex biopsies, according to recent reports. In contrast to fibroblasts, PO-MSCs express CD44, CD90, and stromal cell precursor surface antigen (STRO-1) according to a gene expression study [180]. Scientists can cure POF with PO-MSCs, a novel form of MSC. Oogonial and menstrual blood-derived stromal cells (MenSCs) are two types of stromal cells that originate solely from female ovarian stem cells (OSCs). MenSCs have been used in a number of studies due to their ease of use and lack of invasiveness [157,181]. It suggests that MenSCs can aid in the development of fresh approaches to treating infertility because of their ability for differentiating and flexibility. OSCs, also known as female GSCs, have the potential to form mature follicles when the right conditions are present. Despite debates regarding whether chemotherapy would destroy OSCs, it is still worthwhile to investigate their hidden utility [182,183]. Silvestris et al. demonstrated that the oocyte-like cells obtained by OSC differentiation in vitro, including those from OSCs of women who are in menopause, express markers of meiosis. This ovarian neo-oogenesis model will help develop treatment approaches for female infertility [184,185]. Despite the fact that additional research into these observations is required to clarify OSC function in ovary physiology, clinical research and researchers studying female infertility are currently focusing on OSCs as a novel opportunity to restore ovarian reserve in young women undergoing early ovarian failure and cancer survivors experiencing iatrogenic menopause.

#### 4.2.6. Engineered Stem Cells

Stem cells can be created using gene editing technology to have more sophisticated activities and selectivity compared to their natural qualities. These engineering methods will further enhance the value of next-generation stem cells and their clinical application [186]. By transfecting MSCs with different vectors, it is possible to genetically modify them to improve their phenotype. First, allogeneic stem cells with reduced immunogenicity can be a source of universal donor cells by having human leukocyte antigen genes knocked out [187]. Second, with modified MSCs, focused migration might be noticeably enhanced [188]. Third, in order to improve tissue healing, modified MSCs have the capacity to overexpress cytokines and growth factors. Transfection with microRNA (miR)-21 lentiviral vector, plasmids, or phosphorylated vascular endothelial growth factor (pVEGF) to overexpress certain regulatory factors are two examples of therapies that can be used to improve the health of MSCs through heat shock pretreatment. Although it is still in its infancy, this approach appears to be a novel, safe, and practical treatment for POI. However, before it can be utilized as a potential POI therapeutic, it may require additional in-depth knowledge and preclinical safety testing.

#### 4.2.7. Induced Pluripotent Stem Cells

Induced pluripotent stem cells (iPSCs) were successfully created for the first time in 2006 [189]. Four transcription factors were added by researchers to somatic cells to produce iPSCs. Despite the fact that iPSCs have been linked to a number of potential dangers, such as genetic and epigenetic abnormalities and an increased risk of cancer due to the overexpression of oncogenes like c-Myc [190,191,192], iPSCs still have a significant chance of restoring ovarian function [193,194]. In 2016, it was shown that hiPSCs might be used to create ovarian granulosa like cells (OGLCs). By transplanting these iPSC-derived OGLCs into POI mice markedly increased ovarian tissue, expressed the ovarian granulosa cell markers, raised estradiol levels, and decreased the number of atretic follicles [194]. In 2013, Liu et al. reported that microRNA-17-3P induction and inhibiting vimentin expression can transform hiPSCs into estrogen-sensitive ovarian epithelial like cells (OSE-like cells) [195]. When POI mouse model was transplanted with these OSE-like cells, it resulted in decreased vimentin expression although significantly enhanced the estrogen level and ovarian weight. Kang et al. study reported that the synthesis and secretion of estrogen significantly increased when murine iPSCs were transformed in vitro into functional GC-like cells [196]. According to Zhang et al. the functional GC-like cells derived from rat iPSCs and ESCs express a particular kind of GC receptor; however, endorsing the bona fide nature of these cells as GCs requires further confirmation. Upon coculture with GCs, the ESCs and iPSCs showed matching function, morphological changes and gene expression [197]. Later on, Ling et al. successfully reprogrammed iPSCs in vivo using POI mouse model in 2015, and reported the similar function, morphology and gene expression of iPSCs to that of hESCs [198]. In addition, it was reported that a culture containing Want3a and bone marrow morphogenetic protein 4 (BMP4) can successfully transform iPSCs into germ cells. Even though there are still a lot of problems with transplanting iPSCs, such as trends in tumor evolution and technical problems after transplantation, iPSCs could revolutionize the way disease is handled in biomedical research. Table 1 lists several preclinical studies of stem cell types used to treat POF.

It is evident that stem cells have potential in the treatment of ovarian aging. Nearly all preclinical studies have shown to be promising in POI therapy for almost all stem cells obtained from various sources. However, despite the preclinical benefits of stem cell applications in treating premature ovarian failure and improving ovarian function, practical challenges remain for use in clinical trials.

The foremost issue of major concern is safety. The ovary is the main organ for exercising fertility, and not only affects the health of women but is also very critical to the safety of children. Therefore, the application of stem cells must ensure their safety in the evaluation and treatment of ovarian aging. However, there are some deficiencies in the autologous transplantation of stem cells for clinical application, as they are adult cells and their proliferative and differentiation potential decreases after multiple passages in culture. Furthermore, their differentiation and proliferative capability significantly decrease with an increase in age and aging-related diseases. In addition, immune rejection, subculture, insufficient cell sources and ethical issues of stem cell transplantation is also an important issue that must be addressed before its clinical application.

Next, the efficacy of stem cell transplantation in the assessment and treatment of premature ovarian failure should be explicated, but to accurately predict stem cell efficacy outcomes, valid preclinical models are required, which have been lacking in the context of POI. What is more, stem cells transplantation has also been only limited to preclinical studies. However, humans differ from animal models based on various aspects of biology, genetics, and immunology, which can therefore lead to stem cell transplant failures and unsuccessful transition from animal to human therapy.

Other potential challenges of stem cells for the clinical assessment and therapy of premature ovarian disease cannot be overlooked. For instance, the transition of stem cells from laboratory use to clinical application necessitates validating their biocompatibility with human tissue, which may require further demonstration in preclinical studies. In addition, the clinical translation of stem cell transplantation is also limited due to their unclear therapeutic mechanism in ovarian aging and may require long-term scientific research. Another major concern for several females in developed and underdeveloped countries are due to delayed childbearing and therefore developing POI or going into menopause without completing motherhood. If these females do not bank embryos or oocytes in their earlier life, the only option for starting family are the usage of donated oocytes or adopting a child, which entails enormous costs, psychological implications, and ethical challenges. Therefore, extensive research is in progress for women suffering from POI on how to recruit their remaining follicles, activate them, or find an alternative to develop their reproductive functionality. This review describes various new approaches with positive outcomes in infertility treatment that hold promise for future clinical applications.

### 4.3. Recent Advancement in hiPSC Technology for Restoring Infertility

In the field of biology, some problems are much harder to study than others, and human aging has been considered to be among one of them. The molecular mechanism of human aging is mostly concluded solely based on correlations and their direct experimental exploration is often not realistic. One way to dissect human aging at the molecular level is to study the naturally occurring diseases of premature aging. One most striking and prominent disorder is premature ovarian insufficiency (POI). Due to the complexity and specificity of POI, there are currently no specifically available treatment options for POI [215]. Recently, some stem cells have been used against chemically induced POI models with promising results. However, the existing stem cell therapy is unable in most cases of infertility, especially in couples with the lack of functional gametes and must need donor gametes. Also, it is very important and keen desire of almost all couples to have their own genetically identical child. Recent advancement in stem cell technology increases the probability of in vitro gametogenesis derived from hiPSCs and therefore might provide new therapeutic strategies for infertile couples and successfully overcome the important issues of immune rejection and ethical issues related to the human embryo that was difficult to overcome.

Developing iPSCs from human somatic cells from various organs has advanced greatly, and novel techniques have been used to do so without integrating viral vectors or transgenic sequences [216,217,218]. Individualized regenerative therapy options that restore fertility and/or endocrine function may be made possible by using iPSCs to construct models of the reproductive organs. This would enable novel medication testing. Yamashiro et al. (2020) have recently described a detailed protocol to generate iPSC from somatic cells and later on to differentiate into germ cells and subsequently to oocytes development. During the development of germ cells, mammals are likely to experience certain events and cell-fate transitions, which may affect the normal process of cell formation [219]. However, there is a possibility for critical species differentiation that may take place due to numerous factors. Saitou and Miyauchi (2016) have recognized these factors to vary in accordance with the time scale for a transition of cell fate, developmental synchronicity, signaling, transcription and metabolic features of homologous cell types [220,221]. This therefore entails undertaking a deeper study to learn the mechanism of the development of human germ cells. The mechanism of development of germ cell is also largely unknown, and thus requires detailed investigation. It is also important since it will be helpful in understanding as well as managing many related critical diseases, including infertility as well as genetic and epigenetic offspring disorders. Moreover, the mechanism has been widely studied in other species like mice [31,222], but is already under-studies in the case of human beings.

### 4.3.1. iPSC Differentiation towards Human Germline Cells

The generation of patient-specific stem cells by reprogramming has been almost normal since Takahashi and Yamanaka [223] developed the first line of iPSCs from adult cells. With the help of iPSC technology, it is possible to research the physiopathological and genetic causes of POI in vitro, delineate its characteristics, and examine normal development. By reprogramming stem cells from human somatic cells have been used for disease modeling, and stem cell collections have increased over the past few years. Although iPSCs may be made from any kind of somatic cell, skin fibroblasts are the most common source since they are more widely available. Generating gametes from iPSCs for these patients would be a cutting-edge tactic that would help in determining the specific cause of the disease and ultimately be a great success in developing specific therapy that give hope for parenthood.

Primordial germ cells (PGCs), which are epiblast cells that have been differentiated, are visible at 4 weeks of gestation and migrate along the ectodermal ridges to fill the gonadal ridges at 7 weeks. PGCs actively multiply and start significant nuclear reprogramming when migrating in order to restore their ability to self-renew and reverse their genetic imprint. Germ cells are highly specialized cells that are created by a particular transcriptional mechanism, which controls the comprehensive epigenetic reprogramming of the genome and suppresses somatic destiny [224]. Some important pluripotency-specific genes, such as Blimp 1 (or Prdm1) and Prdm 14, that appear to be involved in mammalian germ cell specification are expressed by cells that differentiate into PGCs [225,226]. Another research finding showed that by using PGCs-specific surface markers such as EpCAM and Integrinα6, they were able to distinguish human PGCs from other cells [227]. The gonadal ridge marks the end of these processes. After several rounds of proliferation, PGCs eventually develop within the sex cords into oogonia or gonocytes of both females and males, respectively.

Due to the complexities of gametogenesis in vivo, modeling in vitro germ cell differentiation will greatly help us in understanding of how developmental processes, including specification, migration, and sex determination, are regulated to transmit genetic information and develop new individuals [45,228]. Currently, researchers have developed PGCs by differentiating iPSCs in vitro, producing viable gametes and progenies in mice. The iPSC capability of developing into germ cells has been reported for both sexes in several species [229,230].

### 4.3.2. Derivation of PGCLCs from Pluripotent Stem Cells

Many of the iPSC-derived germ cells possess the capability of developing into oocyte-like cells. In 2006, Takahashi and Yamanaka [223] for the first time developed pluripotent stem cells by adding four transcription factors to cultured fibroblasts isolated from mice, which dramatically revolutionized cell-based treatment. They also reported that reprogrammed iPSCs were similar to ESCs in terms of appearance, surface marker expression, activity of telomerase, ability to fully differentiate into three lineages, and having a developmentally appropriate karyotype. Since the iPSCs are easily available and derive from adult cells rather than embryos, the iPSCs are preferable to ESCs in the field of regenerative medicine [231]. Additionally, for the reason that patient-specific somatic cells are used to generate iPSCs, there is a lesser chance of immunological refusal. Several investigations have been carried out on the in vitro development of iPSCs into male germ cells, Eguizabal et al. [232] developed the haploid gametes-like cells from differentiation of umbilical cord blood and keratinocytes. In order to establish a differentiation procedure, the culture medium was first added with retinoic acid (RA) for three weeks. They then collected the cells for 2, 3, and 4 further weeks while supplementing them with the recombinant human leukemia inhibitory factor (LIF), forskolin and the inhibitors of CYP26 such as R115866 [232]. In order to xenotransplant human skin cells into the testes of busulfan-therapy generated immunodeficient nude mice, Ramathal et al. [233] cultivated skin cells taken from a patient with azoospermia and infertility in the culture containing RA, BMP8, BMP4 and LIF. They showed that the cells did not develop into germ cell-like cells (GCLCs) outside the tubule, although the transplantation of iPSCs into the seminiferous tubules works. Primordial germ cells (PGCs) give rise to sperm and ova [233]. Human PGCs’ essential regulator was SOX17, according to research by Irie et al. [234], while during specification the BLIMP1 served as a repressor. Therefore, the iPSCs generated from fragile X male and female patients were used in their investigation. According to the research results of Sasaki et al. [227], the hiPSCs in the presence of CHIRON, activin, SCF, BMP4/BMP8, LIF and EGF may differentiate into hPGCs. By using PGC-specific surface markers such as EpCAM and Integrinα6, they were able to distinguish human PGCs from other cells [227]. In other research, by applying standard culture media and/or via an in vivo xenograft approaches, the fibroblast-derived iPSCs were shown to differentiate into gametogenic cells [235]. However, most studies reported that most germ cells derived from the hiPSCs was at the early stages of hPGCLCs and therefore may need to further develop the in vitro condition to enable the differentiation of hPGCLCs into later stages.

### 4.3.3. In Vitro Gametogenesis for Clinical Application

hiPSC-derived oogonia have the potential to serve as a direct precursor state to embryonic oocytes. Rapid advances in the developmental biology the multifaceted cellular regulation have revealed the developmental mechanisms of mouse germ cell, and based on the incorporation of these perceptions, a pioneering protocol has been developed for hiPSC-directed differentiation to reconstitute the ovary. Furthermore, efforts to further improve and optimize the culture system have resulted in the reconfiguration of human germ cell development. Organoids that are more mature and functional will eventually be produced through a novel combination of techniques for tissue engineering and deeper understanding of the human ovary’s development as shown in Figure 5.

Recently, according to Yamashiro et al. [43] and Gell and Clark [29], hiPSCs can be differentiated into oogonia, which gives hope to couples who have insufficient ovarian reserves and are unable to produce viable eggs. After initially differentiating somatic cells into iPSCs through reprogramming, they produced incipient mesoderm-like cells (iMeLCs) in their research in the presence of activin A and Chiron. After that, human primordial germ cell-like cells (hPGCLCs) were produced by plating iMeLCs to aggregate. These hPGCLCs were cultivated for four months with gonadal cells that had been purified from female mouse embryonic ovaries for the purpose of in vitro oogenesis. This provided a new method for the differentiation of germ cells in vitro. The achievement of producing oogonia from human pluripotent stem cells with success gave rise to great optimism and will soon revolutionize reproductive biology. However, there are some key challenges to overcome before they can find their way into clinical applications. Despite the challenging route, recent encouraging results imply that the future is promising. After production has been standardized and the safety of the application has been established, iPSC-technology will eventually be used to treat POI and infertility and many other severe clinical illnesses.

## 5. Prospective of hiPSCs for POI and Infertility

### 5.1. HiPSC-Derived Germ Cells Application in the Field of Regenerative Medicine

The process of differentiating POI patient-specific iPSCs into female germ cells may provide an important disease model for female infertility in vitro. It may be possible to identify abnormalities and decipher the molecular mechanisms associated with human gamete differentiation and maturation during idiopathic premature ovarian insufficiency by comparing the germline differentiation capacity of POI patient-derived hiPSCs to normal hiPSCs. In addition, investigating signaling pathways, epigenetic reprogramming, and transcriptional networks as well as the developmental characteristics of human germ cells will benefit greatly from the differentiation of female germ cells from hiPSCs. Due to the unapproachability of germ cells through early embryonic development and the lack of a suitable experimental model, human germline studies are limited compared to mouse studies [236]. Therefore, the in vitro reconstitution of human germline development is essential, and that may be only possible through the hiPSC differentiation.

### 5.2. Reconstitution of the In Vitro Oogenesis Niche

New insights into the development of germ cells and embryos have been gained through the creation of organoids and stem cell models like hPGCLCs, in vitro-induced gametes, and embryos. We can systematically examine the fundamentals of gametogenesis and embryogenesis by combining organoids, in vivo natural structures, and animal models [237]. Differentiation of in vivo female germ cells depend on the niche possessing Oogenic and somatic cells. The iPSCs differentiation into functional germ cells is extremely important to support the reconstitution of the oogenic niche. However, due to ethical limitations and tumorigenicity, it was not possible to examine whether hPGCLCs transplanted into the ovary could develop into eggs in vivo. Furthermore, with the development of the IVG field and to study hiPSCs role in overcoming infertility, Yamashiro et al. [43] in vitro differentiated hiPSCs into immature germline cells known as PGCLCs, when PGCLCs were routinely achievable in many laboratories [227,238], The importance of this technique lies in the differentiation of early hPGCLCs into more advanced stages such as late PGCs, oogonia and pro-gametogenesis. Yamashiro et al. [219] sorted hPGCLCs from hiPSC-derived aggregates and mixed them with gonadal cells from female mouse embryonic ovaries and generated xenogeneic reconstituted ovaries (xrOvaries). However, the process of hPGCLC-derived oogonia transformation into oocytes and their ultimate competence of fertilization is still in its infancy and is extremely inefficient with less than 10% survival of human germline cells in xrOvaries at the end of the experiment.

Although this new approach of derivation of germ cells from iPSCs is an exciting step forward and may open up windows for germ cells clinical application for infertile individuals. The next most important step in this research field will be the generation of fertilized eggs via IVG. It is presumable that humanized xOvaries may require to complete IVG, and which may involve ovarian somatic cells differentiation from hiPSCs or using human fetal tissue consented to conduct the study. If IVG is successful, it will still face problems and may not be licensed, because the U.S. Department of Human Health and Services does not allow funding of such research projects to produce pre-implanted human embryos through fertilization of sperm and egg cells for research purposes, especially if the resulting embryos may never be used to make babies. Besides, patient specific somatic cells derived iPSCs can possess epigenetic abnormalities or genetic mutation, and therefore be a potent risk of transmitting diseases to the upcoming generations. Previous research has already reported some genetic disorders that have been found in the genome of iPSCs during culturing, reprograming and differentiation [239]. Some countries, such a Japan, which has already made significant progress in this field, have reevaluated the regulated landscape along with the germline genome editing project as an additional technology in the case of single gene mutation, which may lead to the reversal of permanent genetic infertility in the future [240].

## 6. PGC-Derived Exosomes: Future Directions for POI Diagnosis and Therapy

It has been hypothesized that the aging process is connected to a gradual decrease in the number and quality of oocytes in the ovarian cortex’s primordial follicles [37,38]. Pathologically, 1% to 4% of women develop premature ovarian failure (POI), which is characterized by abnormal ovarian function because of the premature depletion of the primordial follicle pool earlier than age 40 [2]. Because assisted reproduction reduces or even depletes ovarian reserve, it can only help to a certain extent, resulting many couples who undergo long-term infertility treatments do not have children. Therefore, for women who are experiencing natural or premature ovarian failure, manipulating these dormant follicles in the ovary may represent a paradigm shift. Recently stem cell transplantation has attracted significant attention and is the most promising approach in regenerative medicine, offering hope for overcoming many major intractable diseases [241]. Although stem cells have been hampered by a number of nonnegligible limitations, including their strong tumorigenicity and immune rejection as well as lack of the understanding of their mechanisms of action [242,243]. Growing evidence shows that ovarian and stem cell transplantation works therapeutically through a paracrine process, leading to a new concept of “cell-free” therapy [244]. Hence, as a replacement for stem cell therapy, stem cell secretions, cytokines and exosomes may be the best choice for clinical application for POI due to their advantages in terms of lack of tumorigenicity, low immunogenicity, low ethical risk and high clinical safety.

Exosomes are extracellular vesicles of endosomal origin measuring 40–150 nm and are carriers of intercellular signaling by encapsulating and transferring microRNAs (miRNAs), mRNAs, and proteins, and other active components to recipient cells [245]. Through their involvement in a variety of biological processes, including inflammation, wound healing, cardiovascular disease, hypertension, cancer, and brain injury, exosomes have been shown to possess intrinsic therapeutic properties for a growing number of diseases [246]. In particular, exosomes have received widespread attention for avoiding the potential risks in stem cell transplantation and having the advantages of extremely low immunogenicity, easy preservation and transportation. Some researchers proposed that stem cell-derived exosomes contribute to POI therapy. Sun and his colleagues found that HucMSC-Exo regulates OGCs apoptosis via the modification of the expressions of gene and protein [247]. Yang et al. [248] showed that the primordial follicles were stimulated by administering HucMSC-Exo into elderly mice aged more than 10 months to activate the development of oocytes. The administered HucMSC-Exo restored fertility and increased the quality and quantity of oocytes. Huang and his colleagues [171] administered hADMSC-Exos into a POI patient’s ovarian granulosa cells and inhibited the apoptosis by downregulating certain genes in the SMAD pathway. In recent years, two important studies reported that chemotherapy-induced ovary damage in mouse models was restored by the application of MSC-exos. Though both studies reported differences in the molecular mechanisms, this emphasizes the regulatory effects of exosomes on the proliferation and apoptosis of granulosa cells [171,249]. These stem cell-derived exosomes open up new directions for POI therapy, but they do not completely solve the problem and are still in the preclinical research stage. Therefore, there is no clear and effective treatment plan to restore the reproductive function of the ovary.

In view of the limitation of current stem cell therapy and lack of an in vitro POI-specific disease model, attempts at identifying various intractable diseases by the early prognostic values of exosomes are underway. It is already known that both normal and diseased cells can produce exosomes [250]. Therefore, here we propose that hPGC-derived exosomes (PGCs-Exos) from healthy and POI-patient specific sources may possess diverse compositions and may serve as a new biomarker in disease diagnosis, mechanisms and treatment measures. It has also been proposed that the plasma level of PGCs-Exo-related miRNAs may also provide a possible therapeutic mechanism and reference for follow-up research for POI treatment. Therefore, studying the changes in exosomal composition in normal or POI patient-specific germ cells will provide a new idea for the treatment of POI. It is hypothesized that PGC-derived exosomes constitute a novel possible cell-free therapeutics that may be used to treat POI.

## 7. Conclusions

Infertility is a global public health concern with serious implications for social well-being, psychological consequences, and an impact on quality of life. In recent decades, the search for cutting-edge treatment methods to help POI patients regain their fertility has been in the spotlight. POI is not only an endocrine disease that describes the continuous decline of ovarian function but also is related to the psychological condition of women, which seriously affects the functions of other organs. However, the regulatory mechanisms of premature ovarian aging remain a mystery, necessitating a greater level of in-depth comprehension in order to fully address this issue and discover more effective therapeutic options.

Recent advances in stem cell technology enable clinicians to perform high-quality interventions to improve ovarian function and restore infertility in POI patients. This review generates insights into a deeper mechanistic understanding of the complex interaction between aging phenotypes and premature ovarian insufficiency and also describes advancements in stem cell technology for POI treatment. However, there is currently no clear and effective treatment option for POI to restore ovarian function and infertility, and only a few preliminary studies have been undertaken. Since the introduction of iPSC technology more than a decade ago, it has now opened up a new direction to develop reproductively functional germ cells and their subsequent oocytes from pluripotent cells with the capability to restore fertility. However, research on this subject is still in the preclinical stage, and there are numerous unanswered issues. Gradually, sophisticated 3D coculture systems are emerging, and through those we will also be able to reconstitute many of the essential interactions between germ cells. Our capacity to model POI using human cells will grow as these methods continue to improve to better simulate in vivo conditions in vitro. As a result, a complete oogenesis process will be generated and may therefore help in accelerating the development of highly effective therapeutic strategies to combat premature ovarian disease and infertility. In this article, we also reviewed the preclinical uses of exosomes derived from different stem cells for POI therapy and proposed hPGC-derived exosomes as a candidate molecule for follow-up research, which might provide a possible novel treatment for this premature ovarian disease.

## Figures and Tables

**Figure 1 cells-11-03713-f001:**
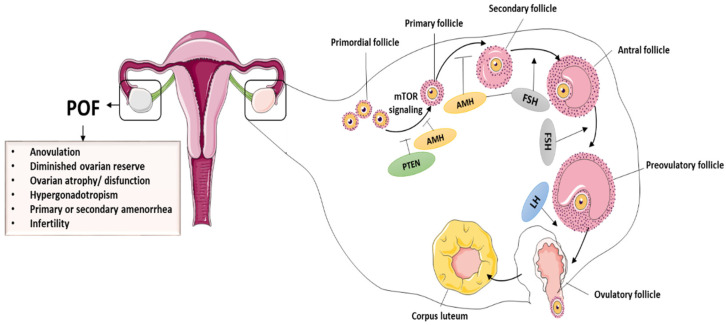
An illustration presenting normal folliculogenesis and symptoms of POF.

**Figure 2 cells-11-03713-f002:**
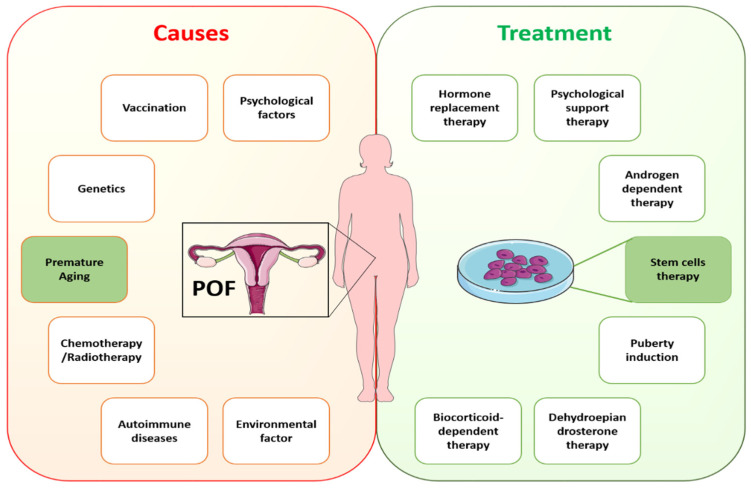
Probable causes of and treatment options for POI.

**Figure 3 cells-11-03713-f003:**
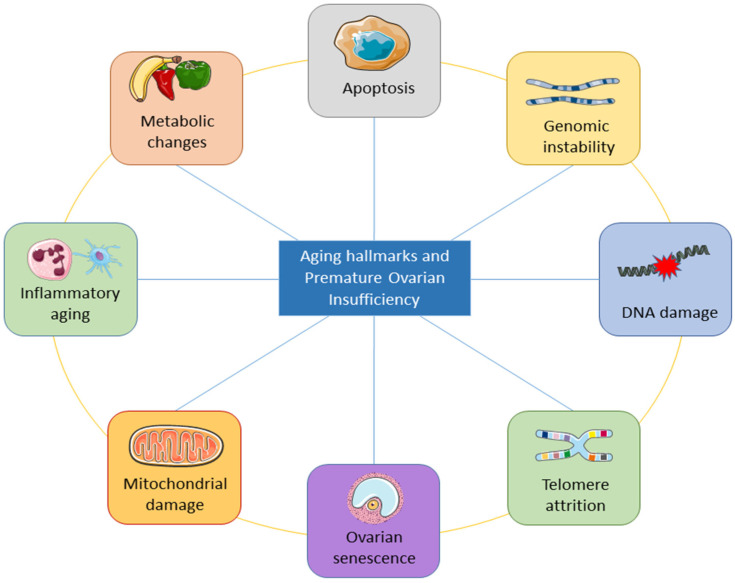
Important aging hallmarks of premature ovarian insufficiency. Inflammatory aging, apoptosis, DNA damage, genomic instability, ovarian senescence, telomere attrition, mitochondrial dysfunction, and metabolic changes seen in POI.

**Figure 4 cells-11-03713-f004:**
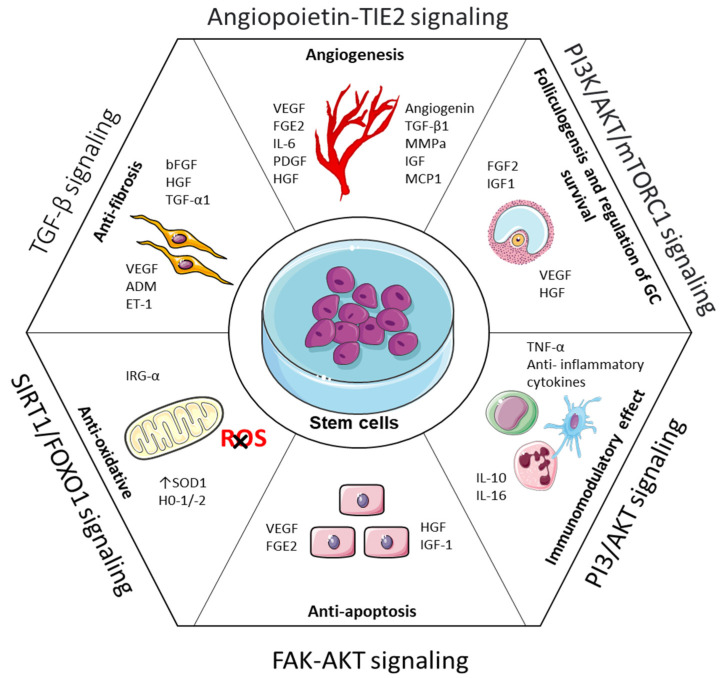
Mechanisms involved in stem cell-based therapies in POF.

**Figure 5 cells-11-03713-f005:**
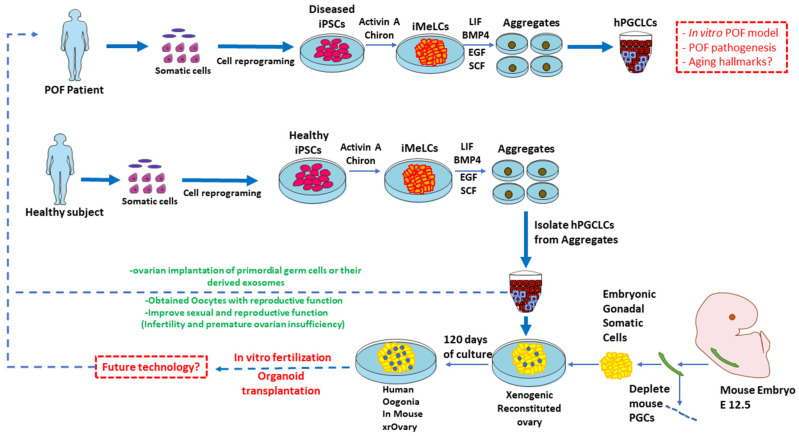
A schematic representation to generate primordial germ cells (PGCs) from POI patient cells for use as an in vitro POI model, and generate human oogonia in xrOvaries from healthy subject somatic cells as a future therapeutic option.

**Table 1 cells-11-03713-t001:** Different types of stem cells used for the treatment of POF in animal models.

Stem Cells	Model	Effect	Reference
Bone Marrow Stromal Cell	Rabbit	Increasing VEGF secretion	[152]
	Mice	The development of fresh primordial follicles	[162]
	Rat	Raising the E2 level, follicle numbers, and ovarian weight	[170]
	Rat	Stopping the apoptosis of GCs	[199]
	Rat	Follicle counts, E2, and AMH levels rising	[200]
Adipose-Derived Stem Cell	Mice	Increasing the proportion of follicles with a healthy structure	[201]
	Mice	Growth of follicles at various phases and ovulation	[155]
	Rat	Rising E2 levels, follicle numbers, and pregnancy rates	[202]
Menstrual Blood-Derived Mesenchymal Stem Cell	Mice	Boost the E2 level, follicle count, and ovarian weight	[156]
	Rat	Boost levels of AMH, E2, and progesterone	[203]
	Mice	Stopping the apoptosis of GCs	[204]
Umbilical Cord Mesenchymal Stem Cell	Murine	Rising ovarian weight, follicle numbers, and AMH levels; increasing follicular expression of Inhibin A and FSHR	[205]
	Rat	Improvement of the endocrine secretion system, folliculogenesis, and inhibition of GCs apoptosis	[176]
	Rat	Estrus cycle recovery, sex hormone levels, and fertility	[206]
	Mice	Decrease in GC apoptosis, rise in sex hormone levels, and increase in follicle count	[150]
	Mice	AMH and E2 levels, ovarian angiogenesis, follicle count, and volume are all increasing	[207]
	Rat	Follicular numbers and E2 levels rising	[208]
Amniotic Fluid Stem Cell	Mice	Follicular atresia prevention while maintaining healthy follicles	[168]
	Mice	Development of new ovarian cells	[209]
	Mice	GC apoptosis and follicular atresia inhibition	[171]
Amnion-Derived Mesenchymal Stem Cell	Rat	GCs’ apoptosis is inhibited, ovarian angiogenesis is increased, and follicular growth is accelerated	[142]
	Rat	Reduction of GC apoptosis, growth of follicles, and elevation of AMH levels	[144]
	Rat	Reducing inflammation	[210]
Placenta Mesenchymal Stem Cell	Mice	GC apoptotic inhibition and elevated E2 levels	[211]
	Mice	GCs’ apoptosis is inhibited, and ovarian function is improved	[212]
	Mice	Rising follicular counts, E2 levels, and AMH levels	[213]
	Mice	Stopping the apoptosis of GCs	[214]
Induced pluripotent stem cells	Mice	Vimentin and fibronectin expression in ovarian tissue was decreased, while ovarian weight and E2 levels were both enhanced	[195]
	Murine	Ovarian tissue expanded, ovarian granulosa cell markers were expressed, estradiol levels rose, and the number of atretic follicles decreased	[194]

## Data Availability

Not applicable.

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
