# Peer review of "Stem Cell-Based Therapeutic Strategies for Premature Ovarian Insufficiency and Infertility: A Focus on Aging"

_cells, 2022, doi:10.3390/cells11233713_

Round 1
Reviewer 1 Report
I enjoyed reading and reviewing this review with novelty in idea and using of new references. ... many paragraphs like 120-150 and 180-200 needs rephrasing agian try to correct many typing errors
Author Response
Response to Reviewer # 1:
Reviewer 1 comments: I enjoyed reading and reviewing this review with novelty in idea and using of new references. ... many paragraphs like 120-150 and 180-200 needs rephrasing agian try to correct many typing errors
Response: Thanks for your appreciation and valuable comments. We wish to express our appreciation to the reviewer for their insightful comments, which have helped us significantly to improve our manuscript. According to the suggestions, paragraphs 120-150 and 180-200 in particular, and the whole manuscript in general have been thoroughly revised and improved its grammar and technical writing, hope now it will be according to the standard of the journal. The final version of the manuscript with highlighted changes is attached.
Reviewer 2 Report
This is a comprehensive and well presented review of the subject, which would be a useful addition to the literature.
The language is generally good, but the whole manuscript needs to be reviewed for fine grammar, punctuation and spelling corrections.
Author Response
Response to reviewer # 2
Reviewer 2 comments: This is a comprehensive and well-presented review of the subject, which would be a useful addition to the literature.
The language is generally good, but the whole manuscript needs to be reviewed for fine grammar, punctuation and spelling corrections.
Response: We would like to thank the reviewer for his appreciation of the work and careful and thorough reading of this manuscript and constructive suggestions. We have thoroughly checked and improved the grammar, spelling corrections, and technical writing of the whole manuscript. Hope now it will be according to the standard of this journal.
Reviewer 3 Report
I read with great interest this comprehensive review on stem cell-based strategies for the treatment of POI and related infertility. The authors performed an excellent work. The manuscript is clear, well written and updated with the most recent literature.
I just have a few minor comments:
- Please define CHR in the sentence "referred by CHR researchers"
- Relating to telomere attrition I suggest to replace with the term "shortening"
- On the clinical use of ovarian stem cells I suggest to update the text with the evidences provided by Silvestris E (In vitro differentiation of human oocyte-like cells from oogonial stem cells: single-cell isolation and molecular characterization, 2018; Ddx4+ Oogonial Stem Cells in Postmenopausal Women's Ovaries: A Controversial, Undefined Role, 2019; In Vitro Generation of Oocytes from Ovarian Stem Cells (OSCs): In Search of Major Evidence, 2019)
Author Response
Response to reviewer # 3
Reviewer 3 comments: I read with great interest this comprehensive review on stem cell-based strategies for the treatment of POI and related infertility. The authors performed an excellent work. The manuscript is clear, well written and updated with the most recent literature.
Response: We are very grateful to the reviewer for their valuable comments and appreciation of the work. We responded to each comment and revised the text accordingly. Our response follows.
I just have a few minor comments:
- Please define CHR in the sentence "referred by CHR researchers"
Response: We appreciate these valuable comments. CHR along with some other abbreviations has now been written in its full form for the first time use.
- Relating to telomere attrition I suggest to replace with the term "shortening"
Response: We have modified it now and replaced the attrition word with shortening.
- On the clinical use of ovarian stem cells I suggest to update the text with the evidences provided by Silvestris E (In vitro differentiation of human oocyte-like cells from oogonial stem cells: single-cell isolation and molecular characterization, 2018; Ddx4+ Oogonial Stem Cells in Postmenopausal Women's Ovaries: A Controversial, Undefined Role, 2019; In Vitro Generation of Oocytes from Ovarian Stem Cells (OSCs): In Search of Major Evidence, 2019)
Response: We appreciate these valuable suggestions. We have further updated the text under the title “4.2.5 Ovarian stem cell” with the relevant evidence provided by Silvestris E in the suggested articles which have also been cited there.